# Is Policy the Necessary or Sufficient Driving Force of Construction and Demolition Waste Recycling Industry Development? Experience from China

**DOI:** 10.3390/ijerph20064936

**Published:** 2023-03-10

**Authors:** Jingru Li, Jinxiao Ji, Jian Zuo, Yi Tan

**Affiliations:** 1Department of Construction Management and Real Estate, College of Civil and Transportation Engineering, Shenzhen University, Shenzhen 518060, China; lijr2000@szu.edu.cn (J.L.);; 2Key Laboratory for Resilient Infrastructures of Coastal Cities, Shenzhen University, Ministry of Education, Shenzhen 518060, China; 3Sino-Australia Joint Research Center in BIM and Smart Construction, Shenzhen University, Shenzhen 518060, China; 4School of Architecture & Built Environment, The University of Adelaide, Adelaide 5005, Australia; 5Entrepreneurship, Commercialisation and Innovation Centre (ECIC), The University of Adelaide, Adelaide 5005, Australia

**Keywords:** construction and demolition waste (CDW), recycling industry, policy strength, event history analysis (EHA), fuzzy set qualitative comparative analysis (fsQCA)

## Abstract

Policies have long been considered the essential driving force in promoting construction and demolition waste (CDW) recycling. However, the policy instruments adopted in different economies have varied greatly, which contributes to the difficulty in quantitative discernment of their effect. This study aims to examine whether the holistic employment of policy measures determines the development of CDW recycling around China. To accurately measure the holistic adoption of CDW policies, this study assessed policy strength via a proposed three-dimensional evaluation model. The spatiotemporal differences in policy strength among the 52 sample cities were further defined using K-means clustering and the Gini coefficient. Next, the driving effect of policy on the initial establishment of CDW recycling industry practices was examined by event history analysis (EHA). Finally, fuzzy set qualitative comparative analysis (fsQCA) was used to analyze the sufficiency and necessity of policy for the initial establishment of CDW recycling practices. The results indicated that the establishment of a first CDW recycling plant is only slightly correlated with policy measures, whereas it is highly correlated with the pilot city and per capita GDP. Furthermore, application of policy is neither a necessary nor sufficient condition for the establishment of a CDW recycling industry facility.

## 1. Introduction

Policies have long been believed by both government authorities and scholars to be the most essential driving force in promoting construction and demolition waste (CDW) recycling [1,2]. Thus, a variety of policy measures have been developed and enacted by global authorities. Regulatory policies are commonly the first to be adopted, followed by economic incentive measures [1,3]. In recent years, information-based measures have gained increased attention since authorities believe they can change the beliefs and behaviors of related stakeholders, with cost efficiency [4]. However, the policy instruments adopted in different economies have varied greatly in their economic, social, and political contexts [5,6]. These variations contribute to the difficulty in quantitative discernment of the actual effect of policy instruments on CDW recycling.

The above situation also appears among Chinese cities. Under pressure from Chinese central government to establish recycling of CDW, municipal governments have responded with great variation [7]. A few cities such as Shenzhen, Qingdao, and Xuchang have made full use of all kinds of policy measures, while many other cities focus only on specific instruments or simply even adopt wait-and-see attitudes. Simultaneously, the CDW recycling industry in China reveals seriously unbalanced states of development [8,9,10]. Correspondingly, it raises the research question: Does the employment of policy measures determine the development status of CDW recycling around China?

Most previous studies focused on qualitative comparisons of CDW recycling measures [5,6,11,12], while quantitative analyses were rarely conducted. Although Li et al. [13] used the policy measures number for quantitative analysis, the manner in which these analyses were designed was neglected, which has a direct effect on their implementation and outcome [14]. In addition to policy tools, other conditions such as population density and economic development may be critical factors for the development of CDW recycling as well [15,16]. These conditions may boost the usage of some policy measures and restrain others. Are policies alone sufficient to motivate the development of the CDW recycling industry? Or do their roles work best together with other conditions? These questions remain unexplored.

Thus, this study aims to examine the actual impact of policy measures on the development of the CDW recycling industry by taking 52 Chinese cities as examples. As shown in Figure 1, this study first accurately measured the usage of CDW recycling policies in each sample city. The next step was to examine whether policy measures have a significant impact on the initial establishment of CDW recycling industry facilities across Chinese cities using event history analysis (EHA). Finally, whether policy measures are necessary or sufficient conditions for the initial establishment of CDW recycling plants was explored using fsQCA.

## 2. Literature Review

### 2.1. The CDW Recycling Policy

Policies are well recognized as an essential driving force to promote CDW recycling in both developed and developing countries [1,17,18]. Three types of policies, including compulsory policy, market-oriented policy, and information-based policy are most commonly employed [19]. Compulsory policy mainly relies on government power to regulate CDW recycling. Complete regulations lay a vital foundation for achievement in CDW recycling [3]. Market-oriented policy encourages enterprises involved in CDW recycling by economic motivation such as improving charge level [20]. Information-based policy involves certificating recycled products or providing publicity and education programs [21].

Different authorities are inclined to employ various mixtures of instruments in consideration of resource limitations or political context [22,23,24]. This disparity may be an essential contributor to the unbalanced CDW recycling development [13]. However, this issue has been overlooked by previous studies, most of which concentrated on comparing the policy mix used in different economies on a qualitative level. Li et al. [11], for instance, compared the policies of CDW recycling adopted in Japan, Germany, and Singapore, and then made recommendations to Chinese authorities. Gálvez-Martos et al. [6] summarized the best practices for the management of CDW and its application in Europe. Aslam et al. [5] analyzed the difference in CDW management policies between China and USA. Tam [25] analyzed the recycling effects of concrete in the construction industry in Australia and Japan and made recommendations for recycling concrete in Australia based on the Japanese experience; Akhtar and Sarmah [26] reviewed CDW recycling in many countries across the world and discovered that compulsory policies and market-oriented regulations are essential in nations with high recycling rates, such as Austria and Japan.

Several studies quantitatively evaluated the effectiveness of policy using System Dynamic Simulation, but were only concerned with special policies in certain cities [27,28,29,30,31]. One of the rare studies that attempted to quantitatively compare the CDW recycling policies adopted by various municipal authorities was reported by [13], which aimed to identify the effective policy measures in the context of China. However, this study only considered the number of policy tools, while the strength and the longitudinal evolution of these CDW policy measures were all overlooked.

### 2.2. Measurement of Policy Strength Using Content Analysis

Content analysis is a widely used qualitative and quantitative research method in the social sciences [32]. This method uses a preponderance of systematic description of the elementary facts and the trends of related topics within text data [19]. Many scholars have employed this method in policy analysis. For instance, Alves and Lee [33] applied content analysis to uncover the aims, governance mechanisms, normative, and targeted areas of the Belt and Road Initiative policy and speeches. Kristianssen et al. [14] analyzed Swedish Vision Zero policy for safety by identifying the policy decision, policy problem, policy goal, and policy measures via comparative content analysis.

Thus, theoretically speaking, this method is capable of identifying and evaluating the manner in which a policy is developed. The critical operation is to propose the evaluation standards. Peng et al. [34] argued that the assessment of a policy’s strength can be conducted by considering policy hierarchy, policy goal, and policy operability. The strength of a policy in China is tightly related to the hierarchy of the issuing departments [19]. The policy made by the higher level of the government institution should be assigned a high score on the index “policy hierarchy” accordingly. If the policy goal is set to address a high priority problem, sufficient resources and support will be allocated [22]. Thus, the high level of a policy goal will receive a high evaluation score. Otherwise, high value is allocated to the policy measures with clear and detailed stipulations which will facilitate the operation and ensure the effectiveness of the policy [34,35]. The values of policy hierarchy, policy goal, and policy operability are positively related to the policy’s effect.

### 2.3. Event History Analysis (EHA)

EHA is a method to explain patterns and causes of the occurrence of an event at a given time [36]. By exploring the similarities and differences encountered by different subjects (e.g., cities), it reveals whether these disparities affect the occurrence or not of an event [37]. With the advantages of accommodating right censoring and modeling the effect of time-varying covariates, this approach has been widely applied to explain the diffusion of techniques or policy. For instance, Raghoo and Shah [38] employed this technique to explore the driver of formation and adoption of the “increasing versus decreasing balance policy” in China. Hossain et al. [15] examined the effect of ownership and location advantages on companies’ decisions about the level of control in international joint ventures by EHA. Other applications included the innovations in archaeological networks [39], recidivism of juvenile offenders [40], employee turnover [41], and the diffusion of educational ideas [42]. However, this method is rarely used in the field of waste management. Indeed, treating the initial establishment of the CDW recycling industry as an “event” can be helpful in deciding whether the policy measures play a determinant role in this event.

### 2.4. Fuzzy-Set Qualitative Comparative Analysis (fsQCA)

The qualitative comparative analysis (QCA) method explores various configurations of antecedent conditions that lead to the occurrence of a particular outcome [43]. Crisp-set Qualitative Comparative Analysis (csQCA), multi-value Qualitative Comparative Analysis (mvQCA), and fuzzy-set Qualitative Comparative Analysis (fsQCA) are the three primary types of QCA [44]. When the variables are discrete, csQCA and mvQCA can be used to solve the binary classification and multi-valued classification problems, respectively, but the loss of information due to discrete variables can raise the problem of contradictory configurations [45]. FsQCA greatly avoids this issue by allowing the outcome and predictor variables to be classified on a fuzzy scale (continuous) with the value of different variables between 0 and 1 [46]. This method is often used to explore multiple pathways to achieve the outcome [47]. For instance, Zhang et al. [48] utilized fsQCA to find that market-developed type, political link type, financial performance type, and state-owned enterprise subsidy type have superior CSR performance from a combination of variables. Marks et al. [47] used the approach to identify three ways for achieving various geographic regional rural water supply continuum functions. It has been used in the fields of enterprise management [49], environment management [50], education [51], and medicine [52], but is less used in the CDW field. FsQCA is innovatively used to explore the antecedent condition configuration for the establishment of the first CDW recycling plant, as it can capture case-complexity information that varies with level or degree during classification [53].

## 3. Research Methods

This study is conducted according to the framework shown in Figure 1.

### 3.1. Sample Cities Selection and Data Collection

Considering the representativeness, a total of 52 cities in mainland China were selected as sample cities. Firstly, 31 provincial capitals and municipalities directly under the central government, which are commonly the economic and political centers of provinces or regions, were included. Secondly, 35 pilot cities specified in the notice on carrying out the pilot work of CDW treatment issued by the Ministry of Housing and Urban-Rural Development in early 2018 were considered. As parts of cities fall into both categories, the total number of sample cities is 52.

The policy documents related to CDW recycling issued by these 52 cities from 2005 to 2019 were collected from the relevant websites (e.g., municipal government websites) and laws and regulations databases. Using “CDW management” and “CDW recycling” as keywords, a total of 252 policy texts were retrieved, including laws and regulations, plans, opinions, methods, notices, and so on. After eliminating unrelated policies by title and content, 221 valid policy texts were used for further analysis.

According to the PEST (political, economic, social and technological) model [54], among political context, economic development, social situation, and technique level, all may have an impact on industry development. Economic development is normally indicated by per capita GDP [55], while social conditions involve many aspects. The population density was found to have an effect on the development of CDW recycling [15,16]. In addition, in the Chinese context, being assigned as pilot city always strengthens the concern of both public and local authorities, which will result in more positive action [56]. Thus, population density and pilot city (recorded as 1, otherwise 0) were incorporated as social condition variables in this study. The technological level is represented by patents in many studies. However, this variable is not considered, due to incomplete records. GDP per capita and population density for 2005-2019 comes from the “Chinese City Statistical Yearbook”, and the pilot city refers to The Ministry of Housing and Urban Rural Development [57].

### 3.2. Measurement of Policy Strength

Instead of counting the policy number, this study measured the policy strength (PS) using policy hierarchy, policy goal, and policy operability. Correspondingly, a three-dimensional evaluation model was established as follows.

In each city, the PS of a policy measure was evaluated using the Equation (1). The policies are classified into compulsory, market-oriented, and information-based measures, marked by characters “c”, “m” or “i”. The PS of a policy keep unchanged until the policy is abolished. At the given time t, the PS values of policy measures in force were accumulated for the same category to calculate the total PS (TPS) as shown by Equation (2).
(1)$${PS}_{j}^{k}=({Pg}_{j}^{k}+{Po}_{j}^{k})\times {Ph}_{j}^{k}\;k=c,m,i$$
(2)$${\mathrm{TPS}}_{t}^{k}=\sum_{j=1}^{N_{t}}{\mathrm{PS}}_{j}^{k}$$
where, PS refers to policy strength; Pg, Po, and Ph represent policy goal, policy operability, and policy hierarchy, respectively; j is the jth policy document; k is the type of policy measure, with c = compulsory measure, m = market-oriented measure, and i = information-based measure; TPS_t_ refers to the total PS of policy at year t; N_t_ represents the policies’ number in force at year t.

The policy goal, issuing department, and the content of three types of measures were identified from the policy documents using content analysis and assigned values in light of the evaluation standards (see Table 1). This standard was adapted from the study of Peng et al. [34] on the basis of interviews with two experts on CDW management.

### 3.3. The Spatiotemporal Disparity Analysis

In this study, the K-means clustering and Gini coefficient were employed to reveal the TPS spatiotemporal difference in sample cities.

#### 3.3.1. K-Means Clustering

The k-means clustering is an unsupervised clustering algorithm [58], which iteratively divides the data into k mutually exclusive categories by Euclidean distance to increase the similarities in clusters and decrease the differences between clusters as possible [59]. This study used it to classify cities in terms of their temporal evolutions of TPS.

An index was also constructed to indicate the temporal increase of PS, as shown in Equation (3).
(3)$${\mathrm{TI}}_{s}^{k}=\frac{\sum_{p=1}^{N_{s}}\left({\mathrm{TPS}}_{p,2019}^{k}-{\;\mathrm{TPS}}_{p,2005}^{k}\right)}{N_{s}}$$
where ${\mathrm{TI}}_{s}^{k}$ is the average increase of ${\mathrm{TPS}}^{k}$ in cluster s; ${\mathrm{TPS}}_{p,2019}^{k}$ and ${\mathrm{TPS}}_{p,2005}^{k}$ are the ${\mathrm{TPS}}^{k}$ value of city p at year 2019 and 2005 respectively; N_s_ refers to the number of cities in cluster s; k is same as Equation (1).

#### 3.3.2. Gini Coefficient

The Gini index is widely utilized to indicate spatial differentiation, e.g., the inequality of education [60], and consumption level [61]. The index ranges from 0 to 1, with higher values indicating higher inequality levels. This study employed it to measure the spatial differentiation of CDW recycling policy. It is calculated using Equation (4).
(4)$${\mathrm{Gini}}^{k}=\frac{\sum_{p=1}^{n}\sum_{q=i}^{n}\left|{\mathrm{TPIE}}_{p}^{k}-{\;\mathrm{TPIE}}_{q}^{k}\right|}{2\mu \;*\;n\left(n-1\right)\;}$$
where ${\mathrm{Gini}}^{k}$ refers to the Gini coefficient; n represents the total number of sample cities; ${\mathrm{TPIE}}_{p}^{k}$ and ${\mathrm{TPIE}}_{q}^{k}$ are the evaluated TPS of city p and q; and μ is the average ${\mathrm{TPIE}}^{k}$ of all sample cities.

### 3.4. EHA

This study used EHA to explore whether the CDW recycling policy has significant impact on the initiation of the CDW recycling industry. The event was defined as the establishment of the first CDW recycling plant. The CDW recycling plant refers to the plant for recycling inert CDW such as concrete, brick, and mortar. The discrete time event history analysis was employed since the data were collected by year. It estimated the log odds of an event occurring, controlling for the accumulated risk related with time [62]. Since the event is binary (occurrence or nonoccurrence), the logistic regression model was used.

A specific “city-year” data set was utilized. Each city appears several times from the time origin (2005 in this study) until the year the event occurred or the year of censoring (i.e., 2019) [63]. The dependent variable in each city was initially set at zero. It would be valued at 1 in the year when an initial CDW recycling enterprise was reportedly operating (the occurrence of the event) in this city. The data was collected by searching the website using “first CDW recycling plant” and “city name”. The independent variables included TPS^c^, TPS^m^, and TPS^i^. They were measured using the method presented in Section 3.2. The economic and social variables were also incorporated. The EHA was conducted using Stata 15.0. When interpreting the coefficients or exponentials of the coefficients, logit estimates are effects on the log-odds scale, and exp (β) shows hazard-odds ratios [20].

### 3.5. FsQCA

Data analysis using fsQCA includes three main steps: data calibration, necessity analysis, and results analysis [64].

#### 3.5.1. Data Calibration

Data calibration is the process of converting the data into fuzzy-set membership values [65]. In this study, the antecedent conditions (i.e., compulsory policy, market-based policy, information-based policy, GDP per capita, population density, and pilot city) were converted into fuzzy set membership values between 0 and 1 [66]. The membership degree of each condition can be set by three anchors of inclusion, exclusion, and crossover thresholds. The crossover (0.5) is the middle point that distinguishes the degree of complete disaffiliation (0) from complete affiliation (1) [67]. The upper 25th percentiles, the 50th percentiles, and the lower 25th percentiles of each variable served as the threshold values for the inclusion, crossover, and exclusion thresholds in this study [68].

#### 3.5.2. Necessity Analysis

Necessity analysis is to determine if a single condition must always exist for an outcome to be present (or absent) [48]. Two metrics, consistency and coverage, are derived in this process. The consistency metric indicates the degree to which each antecedent condition is nearly a perfect subset of the outcome variable [69]. In our case, it shows the extent to which a certain antecedent condition always resulted in the establishment of the first CDW recycling plant. In general, it is assumed that an antecedent condition is necessary for the outcome to occur if its consistency is higher than 0.9 [70]. Coverage is defined as the degree to which each antecedent condition explains the outcome variable [71], including solution coverage, raw coverage, and unique coverage. Solution coverage is the percentage of cases that can be explained by all conditional configurations. Raw coverage represents the percentage of cases that this conditional configuration can explain. Unique coverage shows how many cases can only be described by this combination path [43].

#### 3.5.3. Sufficiency Analysis

The combination’s consistency is used to measure sufficiency, with the minimum acceptable level being 0.75 [70]. Before performing sufficiency analysis, a truth table was calculated which listed all possible combinations (2^6^ = 64). The case frequency and consistency threshold were set to 1 and 0.8, separately, based on the sample size (52 as medium size). The complex, parsimonious, and intermediate solutions were obtained by fsQCA 2.0. The intermediate solution outperforms parsimonious and complex ones because it can balance parsimony and complexity through counterfactual analysis [45]. Therefore, Section 4.4 focused only on the intermediate solution. The parsimonious solution represents the most important condition that cannot be ignored by any solution, called the “core condition”; the condition that appears in the intermediate solution but not in the parsimonious one was called the “edge condition” [51].

## 4. Results

### 4.1. The Statistical Results of TPS Evaluation

The TPS for compulsory policy (TPS^c^), market-oriented policy (TPS^m^), and information-based policy (TPS^i^) during 2005-2019 were respectively calculated for each of 52 cities, yielding a total of 780 values for each type of TPS. The statistic description was tabulated in Table 2. The average TPS of compulsory policy (28.289) was the largest, and that of information-based policy (8.220) was the smallest. During the whole studied period, TPS^c^ always keeps a higher value than TPS^m^ as well as TPS^i^. In addition, the average TPS^c^ soared from 11.884 to 55.807, while the average TPS^m^ hoisted over four times. In 2005, no information-based policy was adopted. However, the average TPS^i^ raised rapidly to 26.496 at the end.

### 4.2. Spatiotemporal Disparity of the CDW Recycling Policy

#### 4.2.1. Temporal Disparity

Roughly seen from the results, three evolution trends could be identified. To accurately divide the evolution trends, K-means cluster analysis was employed using Python 3.8.3 with the scikit-learn package. Since the TPS^i^ in 2005 and 2006 were all zero, the data in these two years were excluded in clustering of TPS^i^. The results of the cluster analysis were presented in Figure 2. The bold black line was the clustering center line, which represents the mean value of this category. The TI for TPS^c^, TPS^m^, and TPS^i^ in each cluster were calculated and tabulated in Table 3.

As indicated in Figure 2 and Table 3, the temporal evolutions of TPS revealed significant differences within the three clusters. Cities in cluster 1 acted fast and positively, keeping the leading position over the entire period. On the contrary, cluster 3 lagged far behind clusters 1 and 2. The average total increase of TPS in cluster 1 was larger than those in cluster 2 and far bigger than those in cluster 3.

#### 4.2.2. Spatial Disparity

Similarly, the spatial variances of TPS^c^, TPS^m^, and TPS^i^ were not negligible. East China, south China, and central China generally acted positively and performed better. Northwest China, north China, northeast China, and southwest China generally behaved inactively, other than several economically developed regions such as Xi’an, Chengdu, and Beijing. The Gini coefficients were presented in Figure 3. The results indicated that the spatial disparity of information-based policy among all cities was extremely high (0.981), far larger than those of the other two categories of policy (0.519 and 0.557) in the primary phase. During the studied period, Gini coefficients showed a significant decline trend for TPS^i^ and slight downward trends for TPS^c^ and TPS^m^. The final Gini coefficients for compulsory policy, economic policy, and information-based policy were 0.361, 0.396, and 0.522, respectively.

### 4.3. Event History Analysis

In 2005, no city owned CDW recycling enterprises, while a total of 46 cities had established their first CDW recycling plants by 2019. Among the 35 pilot cities, 32 cities achieved this goal. EHA was conducted to explore whether the huge differences among TPS had an impact on the start-up of the CDW recycling industry in China. Two models were constructed. Model 1 only included the economic and social variables. Model 2 added three TPS variables. The EHA results were listed in Table 4. 

In model 1, the control variables, GDP per capita, and whether the location was a pilot city, had highly significant effects on the establishment of the first CDW recycling plant. After adding TPS variables in model 2, only TPS of information-based policy showed a significant effect at *p* < 0.1. The two control variables remained significant. Pseudo R^2^ was improved slightly, from 0.1910 to 0.2176.

### 4.4. FsQCA Analysis

#### 4.4.1. Necessity Analysis of Individual Conditions

The necessity analysis of the calibrated antecedent conditions was conducted and shown in Table 5.

The consistency of all the antecedent conditions did not exceed 0.9, indicating that no condition was a necessary one for determining the initial establishment of a recycling plant. Similarly, there were no single necessary conditions for unsuccessful establishment of CDW recycling facilities. However, the consistency of “population density” was rated over 0.8 in importance for unsuccessful establishment of CDW recycling plants, which indicated that low population density was a powerful explanatory factor in individual results.

#### 4.4.2. Configurations Analysis

The results were shown in Table 6. In this study, seven configurations were obtained, with consistency levels for both individual configurations and overall solution values above the minimum acceptable value of 0.75. The seven configurations in Table 6 could be regarded as sufficient combinations of conditions for the initial establishment of a CDW industry plant. In other words, these combinations can all foster the establishment of CDW recycling plants. The consistency values of overall solution and coverage were 0.937 and 0.502, respectively. It meant that 93.7% of cases that met any one of these configuration requirements achieved the successful establishment of a CDW recycling plant, and these seven configurations covered 50.2% of total cases.

The seven configurations could be classified into three contexts, i.e., social context, economic-social context, and governmental-economic-social context. In the social context (see configuration 1), the social conditions played the core role. High population density alone provided sufficient conditions for the development of the CDW recycling industry facilities. The low-level adoption of compulsory and information-based policies did not become an impediment. The absence of economic development or assignment as a pilot city was not an impediment. This result implied that in highly populated regions, the need to save valuable land by deviating CDW from the landfill itself was sufficient to support the operation of the CDW recycling plant. The consistency of this configuration was 0.899. The unique and the raw coverage reached 0.129 and 0.167, separately. The data indicated that 89.9% of the cases that met this condition achieved the successful establishment of the CDW recycling plant. In addition, 16.7% of the successful cases can be explained by this causal path and 12.9% of the cases were explained only by this path. Early in 2009, Xuchang pioneered the first Chinese concessionary operation of the CDW recycling plant even without the enactment of related policy measures.

In the economic-social context, the combination of economic and social conditions supported the establishment of the CDW recycling industry even without the strong promotion of related policies. Of these, social conditions (population density and being a pilot city) played a critical role and economic conditions filled a supplemental role. If the economically developed and densely populated cities were selected as pilot cities, they were likely to establish CDW recycling plants. Of the plant-established cases, 4.9% could be explained via this causal pathway, and 4.2% of the cases could only be explained by this pathway. The rather high consistency revealed that the case cities that fit this configuration all achieved this aim. These cities include Changzhou and Dongguan.

In the third context, including the last five configurations, policy instruments definitely promoted the set-up of CDW recycling plants, but all in conjunction with other economic or social conditions. In configurations 3, 4, and 5, only one kind of policy measure was extensively employed, but these cases successfully initiated CDW recycling plants only if the social conditions (dense populations or being selected as pilot cities) were met simultaneously. Nevertheless, the cases uniquely explained by these three conditions were limited. In configuration 6, both economic and informational measures were widely taken, which, however, needed the auxiliary effect of certain social conditions to motivate the establishment of the plant. Even with all three kinds of policy instruments intensively employed in configuration 7, the supplementary economic and social environments were still needed to construct sufficient conditions for CDW recycling industry development. It was noticeable that this configuration uniquely explained 15.5% of the cases.

## 5. Discussion

In this study, we proposed a three-dimensional evaluation model to measure the TPS from policy hierarchy, policy goal, and policy operability. In addition to the number of policies, the way they were designed was considered too. This evaluation provided more accurate descriptions of the power and effects of CDW policy mix in each city, demonstrating the authentic resolution of authority to motivate CDW recycling.

The results revealed that compulsory policy, market-based policy, and information-based policy all maintained increment from 2005 to 2019, but with great differences. To quantitatively determine the spatial-temporal differences of the TPS in the sample cities, k-means clustering and the Gini coefficient were applied. With the k-means clustering method, cities with similar temporal trends were clustered together based on Euclidean distance. The clustering clearly identified three trends—fast, moderate, and slight increments of TPS—for all three types of policy. This result means these sample cities had paid extensive attention to addressing this problem. The Gini coefficient showed that the spatial disparity remained noteworthy despite the declining trend. The active cities were mainly located in south China, east China, and central China, the most economically developed areas. This conclusion is consistent with Yang and Zhou [72]. Economic development usually relies on extensive infrastructure construction and therefore generates greater pressure for CDW disposal. To tackle this issue, the municipal governments in those regions have stronger motivation to comprehensively and effectively design CDW recycling policy measures.

Did the enormous diversity in TPS of policy measures contribute to the variety of stages in the establishment of the CDW recycling industry in each city? EHA was utilized to address this problem. From the EHA results, it was found that only the TPS of information-based policy significantly influenced the initiation of the CDW recycling industry, while coercive and economic policy did not. This result was partly echoed by the Li et al. [13] study, in which an information-based instrument (Green Product Label) was identified as the most salient factor in the CDW recycling industry development in the Chinese context. The EHA results meant that those cities emphasizing the information-based policy were more likely to establish the CDW recycling plant. The underlying reason may be that the CDW recycling enterprises tend to trust open and communicative governments. Compared with those governments preferring coerciveness, the governments advocating information-based measures are easier to communicate and cooperate with. Recycling enterprises, after all, need grand support from governments for their raw material supply, operation, and product promotion [17].

Unexpectedly, the policies did not act as a strong driver for CDW recycling industry development according to EHA results. Economic development level and pilot city status seemed to play critical roles in the EHA results. The result implied that the CDW recycling industry can be developed under suitable market or social conditions even without pressure from governments. It was worth noting that becoming a pilot city was a particularly significant determiner in establishing the CDW recycling industry according to the hazard-odds ratios ($e^{7.1951}$). This conclusion is echoed by Cao et al. [73]. They found that the urban green total factor productivity increased by about 1.24% on average after becoming pilot cities. Population density, however, did not show a significant effect, which contrasted with previous studies [15,16].

FsQCA results, however, revealed the different mechanisms of the political, economic, and social factors on motivation for CDW recycling. First, it indicated that the absence of any single condition would not cause the failure of establishment of the CDW recycling industry. In addition, high population density was sufficient to motivate the set-up of the first CDW recycling plant alone, requiring minor policy support. Indeed, in several densely populated cities such as Handan and Xuchang, the first CDW recycling company was established without strong support from policy measures. This conclusion responds to the above implication from EHA results. In contrast, policy measures cannot play this role alone. They need the cooperation of economic or social condition whenever some or all kinds of policy instruments are adopted. To our knowledge, these conclusions were never proposed by previous studies.

## 6. Conclusions

Are policies the essential driving force to promote CDW recycling? Has the spatiotemporal disparity of policy adoption determined the developmental stages of the CDW recycling industry in Chinese cities? To answer these questions, this study innovatively analyzed data on 52 representative Chinese cities via a combination of content analysis, clustering analysis, Gini coefficient, EHA, and fsQCA. The first novelty is that the policy strength (TPS) adopted from 2005–2019 was extensively measured using our proposed three-dimensional evaluation model. Then, the spatiotemporal disparity of policy efficacies was analyzed via k-means clustering analysis and Gini coefficient. Another novelty lies in the fact that not only individual variables but also the combination of multiple variables in the initial establishment of the CDW recycling plants were analyzed using EHA and fsQCA, respectively.

The results revealed that the strengths of compulsory policy, market-oriented policy, and information-based policy all increased across the sample cities within the studied period, indicating that policies were receiving more attention and were being highlighted by local authorities, thereby motivating development of CDW recycling. However, the significantly varied growth-rates and high-level Gini coefficients revealed the spatiotemporal difference of TPS cannot be ignored. According to the EHA results, only the relationship between information-based policy and the establishment of the first CDW recycling plant had a statistically significant effect at *p* < 0.1 level. The fsQCA results further demonstrated that policy alone was not sufficient to promote the development of the CDW recycling industry; it worked only in combination with economic and social conditions. In contrast, the establishment of a CDW recycling plant can be facilitated by social conditions (dense population) alone or combined with economic conditions.

These results provided valuable references for authorities to enhance CDW recycling in the context of China or other, similar regions. Municipal governments should make policy flexibly, considering the practical context rather than simply policymaking without context. In developing and strengthening policy measures, the focus should be on whether the prevailing social or market conditions at that moment are lasting.

This study has certain limitations as well. First, we did not take technological factors into account due to data unavailability. Second, although 52 representative cities were considered in this study, there are still numerous other cities remaining to be explored. We hope to extend the number of cities in a future study.

## Figures and Tables

**Figure 1 ijerph-20-04936-f001:**
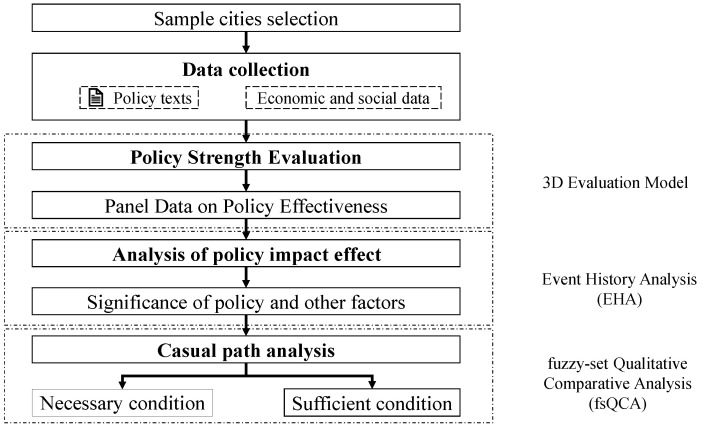
The framework of this study.

**Figure 2 ijerph-20-04936-f002:**
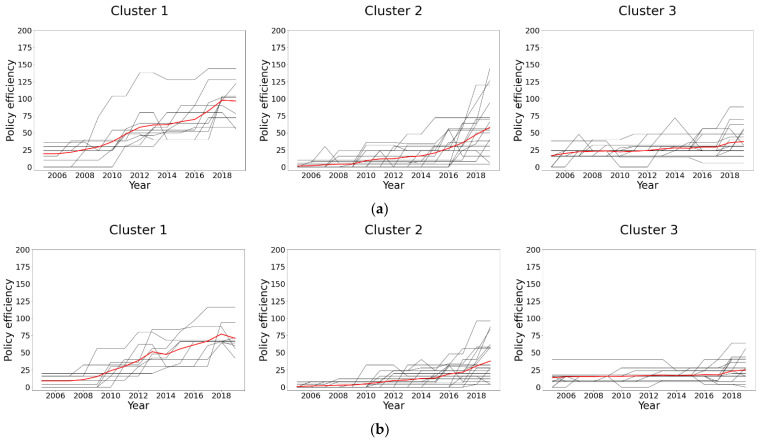
The clustering analysis of TPS. (**a**) Compulsory policy. (**b**) Market-based policy. (**c**) Information-based policy.

**Figure 3 ijerph-20-04936-f003:**
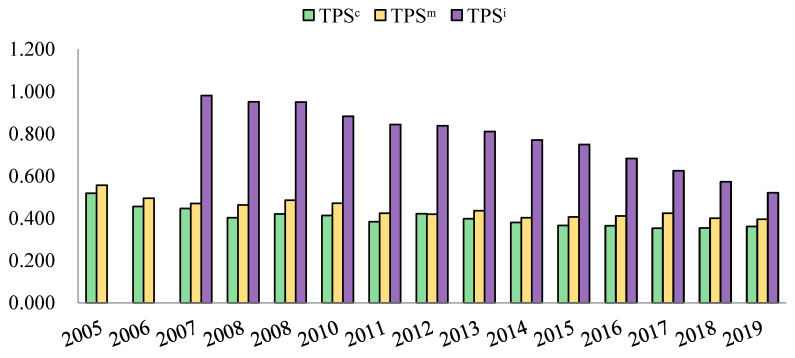
Gini coefficients in terms of TPS^c^, TPS^m^, and TPS^i^.

**Table 1 ijerph-20-04936-t001:** The evaluation standards of three indicators.

Indicators	Evaluation Standards	Value
Policy hierarchy	Issued by the Municipal People’s Congress or its Standing Committee	5
	Issued by the municipal government	4
	Jointly issued by 3 or more departments	3
	Jointly issued by two departments	2
	Issued by a single department	1
Policy goal	Aim to promote the continuous progress of the circular economy	5
	Aim to promote the CDW recycling	3
	Aim to promote general CDW management	1
Policy operability	Presenting requirements on CDW recycling with detailed and clear rules or guidelines, e.g., reward or punishment standards	5
	Presenting requirements on CDW recycling but without detail and clear rules or guideline	3
	Just mention to promotion of CDW recycling but without any detailed requirement	1

**Table 2 ijerph-20-04936-t002:** The statistical results of TPS.

Category	Year	Obs	Mean	SD	Minimum	Maximum
TPS^c^	Total	780	28.289	25.791	0	144
	2019	52	55.807	37.014	4	144
	2005	52	11.884	11.351	0	38
TPS^m^	Total	780	18.879	19.307	0	116
	2019	52	37.884	27.520	0	116
	2005	52	8.000	8.713	0	40
TPS^i^	Total	780	8.220	16.987	0	94
	2019	52	27.653	26.496	0	94
	2005	52	0.000	0.000	0	0

**Table 3 ijerph-20-04936-t003:** The TI index values of three clusters.

Index	Cluster 1	Cluster 2	Cluster 3
TI^c^	77.2	56.4	20.7
TI^m^	61.8	36.8	11.4
TI^i^	68.0	58.8	17.0

c = compulsory measure, m = market-oriented measure, and i = information-based measure.

**Table 4 ijerph-20-04936-t004:** EHA results.

Variables	Model 1	Model 2
GDP per capita	1.0000 *** (6.35 × 10^−6^)	1.0000 *** (7.14 × 10^−6^)
Population density	1.0001(0.0004)	0.9998 (0.0005)
Pilot city	10.3451 *** (5.2423)	7.1951 *** (3.9166)
TPS^c^		0.9925 (0.0186)
TPS^m^		1.0082 (0.0262)
TPS^i^		1.0346 * (0.0194)
cons	0.0110 *** (0.0051)	0.0153 (0.0082)
Number of Obs	638	638
LR chi2	57.12	65.06
Log likelihood	−120.9372	−116.9671
Prob > F	0.0000	0.0000
Pseudo R^2^	0.1910	0.2176
AIC	249.8745	247.9431

* *p* < 0.1; *** *p* < 0.01; St. Err is in bracket.

**Table 5 ijerph-20-04936-t005:** Analysis of necessity for establishment of the first recycling plant.

Condition		Establishment of the First Recycling Plant		No Establishment of any First Recycling Plant	
		Consistency	Coverage	Consistency	Coverage
Compulsory policy	High	0.509	0.777	0.486	0.223
	Low	0.492	0.761	0.514	0.239
Market-based policy	High	0.476	0.769	0.478	0.231
	Low	0.524	0.770	0.523	0.230
Information-based policy	High	0.487	0.786	0.443	0.214
	Low	0.513	0.754	0.557	0.246
GDP per capita	High	0.480	0.748	0.538	0.252
	Low	0.520	0.790	0.462	0.210
Population density	High	0.618	0.928	0.161	0.072
	Low	0.382	0.603	0.839	0.397
Pilot city	High	0.300	0.750	0.333	0.250
	Low	0.700	0.778	0.667	0.222

**Table 6 ijerph-20-04936-t006:** Configuration analysis of the first CDW recycling plant.

Condition Configuration	1	2	3	4	5	6	7
	Social Context	Economic-Social Context	Governmental-Economic-Social Context				
Compulsory policy	⊗	⨂	●	⨂	⨂		●
Market-based policy		⨂	⨂	●	⨂	●	●
Information-based policy	⨂	⨂	⨂	⨂	●	●	●
GDP per capita	⨂	●			⨂	⨂	●
Population density	●	●	⨂	●	●	●	●
Pilot city	⨂	●	●	⨂	●	⨂	
Consistency	0.899	0.995	1.000	0.899	0.984	0.945	0.902
Raw coverage	0.167	0.049	0.038	0.060	0.016	0.082	0.178
Unique coverage	0.129	0.042	0.032	0.021	0.011	0.059	0.155
Solution coverage	0.502						
Solution consistency	0.937						

Note: ● and ⨂ denote, respectively, the presence and absence of conditions; large and small circles denote, respectively, core and edge conditions; blanks represent conditions that may or may not exist.

## Data Availability

We have participated sufficiently in work to take public responsibility for the appropriateness of the collection, analysis, and interpretation of the data.

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
