# Peer review of "Is Policy the Necessary or Sufficient Driving Force of Construction and Demolition Waste Recycling Industry Development? Experience from China"

_ijerph, 2023, doi:10.3390/ijerph20064936_

Round 1
Reviewer 1 Report
Revised the text to improve the expression
Reviewer 2 Report
The article is a very interesting approach on the real effects of policies on CDW recycling in China. The manuscript is well written and the methodology is well defined. As suggestions for improving the article, the following considerations are made:
1. The literature review is very brief. It is recommended that authors seek more information, especially on CDW recycling policies.
2. It is suggested that the authors better discuss the results obtained, presenting the importance of each analysis performed. For example, it wasn't clear to me what the clusters meant and how they were separated.
3. Conclusions are too long. It is suggested that the authors rewrite it in a shorter form.
Reviewer 3 Report
Very valuable study regarding problem of worldwide importance. The results are clearly presented, small improvements should be considered (e.g. Figure 2. The clustering analysis of TPS should be improved because it is illegible a little)
